# Eco-Friendly Edible Packaging Systems Based on Live-*Lactobacillus kefiri* MM5 for the Control of *Listeria monocytogenes* in Fresh Vegetables

**DOI:** 10.3390/foods11172632

**Published:** 2022-08-30

**Authors:** Ramona Iseppi, Chiara Zurlini, Ilaria Maria Cigognini, Mariarosaria Cannavacciuolo, Carla Sabia, Patrizia Messi

**Affiliations:** 1Department of Life Sciences, University of Modena and Reggio Emilia, Via G. Campi 287, 41125 Modena, Italy; 2SSICA-Stazione Sperimentale per l’Industria delle Conserve Alimentari, Viale F. Tanara 31/A, 43121 Parma, Italy

**Keywords:** *Listeria monocytogenes*, lactic acid bacteria, polysaccharides-based films and coatings, protein-based films and coatings, raw vegetables

## Abstract

To meet consumer requirements for high quality food free of chemical additives, according to the principles of sustainability and respect for the environment, new “green” packaging solutions have been explored. The antibacterial activity of edible bioactive films and coatings, based on biomolecules from processing by-products and biomasses, added with the bacteriocin producer *Lactobacillus kefiri* MM5, has been determined in vegetables against *L. monocytogenes* NCTC 10888 (i) “in vitro” by a modified agar diffusion assay and (ii) “on food” during storage of artificially contaminated raw vegetable samples, after application of active films and coatings. Both polysaccharides-based and proteins-based films and coatings showed excellent antilisterial activity, especially at 10 and 20 days. Protein-based films displayed a strong activity against *L. monocytogenes* in carrots and zucchini samples (*p* < 0.0001). After 10 days, both polysaccharide-based and protein-based films demonstrated more enhanced activity than coatings towards the pathogen. These edible active packagings containing live probiotics can be used both to preserve the safety of fresh vegetables and to deliver a beneficial probiotic bacterial strain. The edible ingredients used for the formulation of both films and coatings are easily available, at low cost and environmental impact.

## 1. Introduction

The current trend in lifestyle and food consumption has shown that food preservation must be directly related to novel methods of packaging, including active, intelligent, and edible systems [1]. To create innovative and advanced green packaging, it becomes extremely essential to apply sustainable principles and methods aiming to the protection of natural resources and environment, reducing the use of chemicals and synthetic plastics for food preservation, notably for highly perishable fresh products, such as vegetables. Recently, to extending the post-harvest quality of fresh cut fruits and vegetables, polysaccharide-based [2,3,4,5] or lipid-based materials [6] are used to develop antimicrobial edible coatings and films [7,8,9,10]. Moreover, to meet the consumers’ requirements for high quality and chemical additive free foods, the use of Lactic Acid Bacteria (LAB) and/or their exoproducts (bacteriocins) has been extensively investigated and are widely used in the food industry as well [11]. LAB, microorganisms generally recognized as safe (GRAS) by the United States Food and Drug Administration (FDA) and Qualified Presumption of Safety (QPS) by the European Food Safety Authority (EFSA), are considered as safe alternatives to chemicals, which, in the long-term, may have adverse impact on human health. Bacteriocins are ribosomal synthesized peptides or proteins able to inhibit or kill other related (narrow spectrum) or unrelated (broad spectrum) microorganisms [12]. The antibacterial activity of bacteriocins is due to their interaction with the bacterial cell surface and cell membrane, with cell permeabilization and pore formation as major mechanisms of action [13,14]. LAB are microorganisms with known probiotic properties able to inhibit many microorganisms and cause the production of antimicrobial compounds, such as organic acid, bacteriocins, bacteriocin-like inhibitory substances (BLIS), and other metabolites [15,16,17]. LAB can inhibit *Escherichia coli* O157:H7, *Pseudomonas aeruginosa*, *Salmonella* spp. [18,19,20], and *Listeria monocytogenes*, which are the main foodborne pathogens causing several human diseases [21]. Regarding the microbial risk, attention will be directed to *L. monocytogenes*, a foodborne pathogen associated with a mortality up to 30%, that recently caused outbreaks in European countries. As pointed out in the report of the EFSA and the European Centre for Disease Prevention and Control, in 2019, this foodborne pathogen caused 2621 confirmed invasive human illness, with an EU notification rate of 0.46 cases per 100,000 population in 36 European countries. Listeriosis is one of the most serious food-borne diseases under EU surveillance, with a growing lethality (17.6%) in 2019, compared with 2018 and 2017 (13.6% and 15.6%, respectively) [22]. *L. monocytogenes* primarily affects immuno-compromised individuals and pregnant women [23], and has become an important cause of human food-borne infections worldwide. Being a ubiquitous microorganism, it is frequently found in raw materials, and this explains its presence in many foods. The presence of *L. monocytogenes* on cut and minimally processed vegetables intended for human consumption has been clearly demonstrated in many countries, and the role these vegetables play as vehicles for human infection is now well known [24,25]. On the contrary, the involvement of whole vegetables in the epidemiology of listeriosis is less documented. A recent systematic literature review, which collected the data obtained in numerous studies on the growth and/or survival of *L. monocytogenes* on artificially contaminated whole plant products [26], shows that in many cases the pathogen was able to grow reaching high microbial loads. These products in nature can be contaminated in the production stage and due to hygienically inappropriate handling by workers, but there are many factors that could then favor its development during the shelf-life of the product. Difference in survival and growth of the pathogen before consumption may be due to temperature, humidity, and, in the case of whole plants, the surface topology. The rougher the surface, the better it allows bacteria to adhere and, over time, to form biofilm, a structure that also represents a problem for sanitization in the food sector. The three vegetables chosen for the study are very different not only in surface topology, but also for water and nutrient content. They are frequently used raw in many RTE foods and also widely represented in the Mediterranean diet, and could, as others, be a concern in *L. monocytogenes* contamination. *L. monocytogenes* is also a psychrotrophic bacterium capable of surviving and proliferating at typical refrigeration temperatures, and this feature makes its control in raw foods extremely difficult [27,28,29]. Nowadays, much attention is paid to leading a healthy diet, and new food trends, such as the addition of vegetables to fruit to prepare smoothies and extracts, has led to a considerable consumption of raw vegetables. Furthermore, in these products consumed raw, most of the nutrients remain unaltered, especially those present in the peel, such as lutein in zucchini, lycopene in tomatoes, and beta-carotene in carrots. These compounds are fundamental nutritional principles for our psycho-physical well-being, able to counteract the production of free radicals, responsible for cellular aging. In the present study, these three types of vegetables have been employed to evaluate the antibacterial activity of new packaging systems. Edible bioactive films and coatings (alginate and xanthan gum), based on biomolecules from processing by-products and biomasses, have been added with a viable Lactic Acid Bacteria (LAB), *Lactobacillus kefiri* MM5, strain chosen for the capability to secrete a bacteriocin active against *L. monocytogenes*. The capability to produce the anti-*Listeria* compound by this living LAB strain entrapped in the packaging materials has been determined: (i) “in vitro” by a modified agar diffusion assay, to confirm the retention of the antibacterial activity after the treatments employed to prepare film and coating, and (ii) “on food” during chilled storage of artificially contaminated raw vegetable samples after application of films and coatings.

## 2. Materials and Methods

### 2.1. Materials for Films and Coatings Preparation

Sodium alginate purchased from Sigma Aldrich United Kingdom, was used as a major films and coatings-forming component for the preparation of polysaccharides-based films and coatings, while pea proteins were obtained from industrial processing residues of peas. These by-products, principally constituted by non-compliant seeds of peas, empty pods and a mixture of leaves and stems, have been subjected to an extraction process of proteins, developed and set up by SSICA in a specific Italian patent (SSICA Italian Industrial Invention Patent #.1,399,500) [30]. The protein content was 79.6 ± 0.4%. The determination of the protein content is based on the Kjeldhal method [31,32,33,34]. Glycerol, purchased from Sigma Aldrich, Germany, was added as plasticizer in all the formulations for films and coatings. Xanthan-Gum, purchased from A.C.E.F., Italy (Satiaxane CX 930, Xanthan Gum E415), was used as a thickener in protein coating.

### 2.2. Film Characterization

#### 2.2.1. Thickness

Film thickness was measured by using a digital micrometer (Digital Micrometer QuantuMike IP65, Mitutoyo, Kanagawa, Japan) with 0.001 mm accuracy. Measurements were made in at least ten random locations of each film to obtain the average thickness representing the sample.

#### 2.2.2. Barrier Properties

Water Vapor Transmission Rate (WVTR) was measured using the TotalPerm permeabilimeter (ExtraSolution, Lucca, Italy) instrument, at 23 °C and 10.00% relative humidity according to ASTM F 1249-20 [35]. The sample films were cut into a circle of 4 cm diameter and the test area was 2 cm^2^. Measurements were carried out in quadruplicate. Oxygen Transmission Rate (OTR) was measured using the PERMEO2 permeabilimeter (ExtraSolution, Lucca, Italy) instrument, using 2 cm^2^ samples of the examined films according to ASTM F 1927-20 [36]. Permeability measurements were conducted at 23 °C and 10% RH. Three samples were tested for each type of film, and average results of permeability (expressed as the volume of permeant passing through a film, per unit area and time) are presented. 

### 2.3. Active Films and Coatings Preparation

#### 2.3.1. Bacterial Strains

*Lactobacillus kefiri* MM5, the probiotic strain used in this study to add in both types of films and coatings, was previously isolated in our laboratory from commercial kefir grains in De Man–Rogosa–Sharpe agar (MRS agar, bioMérieux, Milan, Italy). The strain was identified by matrix-assisted laser desorption ionization (MALDI) time-of-flight mass spectrometry (TOF/MS). *L. monocytogenes* NCTC 10888 was used as contaminant in the “on food” study. Both strains were stored until required in phosphate-buffered saline (PBS; 8 g NaCl, 0.2 g KCl, 2.9 g Na_2_HPO_4_·12H_2_O, 0.2 g KH_2_PO_4_ with 1 L of distilled water) supplemented with 20% (*v*/*v*) glycerine at −80 °C.

#### 2.3.2. Anti-Listeria Activity Determination

The anti-*Listeria* activity of *L. kefiri* MM5 was determined by an agar spot assay using *L. monocytogenes* NCTC 10888 as indicator strain. Overnight culture of the *L. kefiri* MM5 (5 µL) was spotted onto Tryptic Soy agar (TSA, Oxoid, Milan, Italy) plates previously spreaded with 10^7^ CFU (Colony Forming Unity) of the indicator. The anti-*Listeria* activity was evaluated after 24 h of incubation at room temperature and at 4 °C and quantified by a clear zone of inhibition in the indicator growth around the spot of the producer.

#### 2.3.3. Preparation of Cultures for “On Food” Studies

*L. kefiri* MM5 cells were cultured in 100 mL MRS broth under anaerobic conditions at 37 °C for 24 h. The cells were harvested by centrifugation at 3000× *g* for 20 min. The supernatant was removed, while the pellets were washed twice with phosphate buffer saline (PBS) at pH 7 and re-suspended in an appropriate volume of PBS. The cell suspensions were directly used for the preparation of films and coatings.

### 2.4. Preparation of Polysaccharides-Based Active Films (PoAF) and Coatings (PoAC)

#### 2.4.1. Polysaccharides-Based Active Films (PoAF)

The polysaccharides films were produced according to the methods published by Olivasa and Barbosa-Canovas, with modification [37]. The film-forming solution was prepared in double deionized water containing 2.00% (*w*/*v*) sodium alginate, 1.50% (*w*/*v*) glycerol. The mixture was stirred for some hours at room temperature using a magnetic stirrer until the ingredients were completely dissolved in solution. Successively, polysaccharides-based films were produced by using casting techniques. More in detail, polysaccharides-based films were casted by pouring 50 mL of the prepared solution into 12 cm side square petri dishes and allowed to dry overnight at room temperature under laminar flow hood. For the preparation of polysaccharides-based active films (PoAF) with *L. kefiri* MM5 viable cells, the microorganisms were added to the solution, once that all sodium alginate and glycerol were dissolved, at room temperature and then the procedure of film formation has been the same as normal standard polysaccharides-based film by casting technique, as described above. The microorganisms were added to the solution in a quantity equal to 3.00% (*w*/*w*) of the final solution.

#### 2.4.2. Polysaccharides-Based Active Coatings (PoAC)

The polysaccharides-based coating has been prepared by dissolving 6.00% (*w*/*v*) sodium alginate in double deionized water at room temperature. For the polysaccharides-based active coatings with *L. kefiri* MM5 viable cells (PoAC), the microorganisms were added to the solution, once all sodium alginate was dissolved, at room temperature. The microorganisms were added to the solution in a quantity equal to 3.00% (*w*/*w*) of the final solution.

### 2.5. Preparation of Protein-Based Active Films (PrAF) and Coatings (PrAC)

#### 2.5.1. Protein-Based ACTIVE Films (PrAF)

The protein films were produced according to the methods published by Bamdad et al., with modification [38]. The film-forming solution was prepared in double deionized water containing 6.25% (*w*/*v*) pea protein and 3.12% (*w*/*v*) glycerol. The pH of the solution was measured and checked in order to have a value of pH of about 7.20, that is the value of pH at which the solubility of pea proteins is maximum. In general, the starting pH was about 6.9 and it was corrected to 7.20 with NaOH. The mixture was stirred for 1 h at room temperature using a magnetic stirrer, the pea proteins presented a good solubility in water and dissolved immediately in water. After the solubilization, the pea proteins were denatured by heating the solution at 70 °C (with bain-marie) for 20 min. After the denaturation, the solution was kept cooling at room temperature and then protein films were produced by using casting technique. More in details, proteins films were casted by pouring in a first trial 5 mL of the prepared solution into 5.50 cm diameter petri dishes and in a second trial by pouring 30 mL of the prepared solution into 12 cm side square petri dishes. In all trials the petri dishes were allowed to dry overnight at room temperature under laminar flow hood. For the preparation of protein-based active films with *L. kefiri* MM5 viable cells (PrAF), the microorganisms were added to the solution, after the denaturation and once the solution was cooled down. Then the procedure of film formation has been the same as normal standard proteins film by casting technique, as described above. The microorganisms were added to the solution at a temperature of about 30 °C in a quantity equal to 3.00% (*w*/*w*) of the final solution.

#### 2.5.2. Protein-Based Active Coatings (PrAC)

The protein-based coating has been prepared first by dissolving 6.25% (*w*/*v*) pea protein and 3.12% (*w*/*v*) glycerol in double deionized water. As for the films, and also for the coating, the pH was measured and checked to 7.20, and the procedure was still the same, with the phase of solubilization of 1 h and the phase of denaturation at 70 °C for 20 min. After denaturation, and once the solution was cooled down, X-Gum was finally added as thickener, in a quantity equal to 0.50% (*w*/*w*). For the protein-based active coatings with *L. kefiri* MM5 viable cells (PrAC), the microorganisms were added to the solution after the check of pH, the solubilization and the denaturation, when the temperature of the solution was about room temperature. The microorganisms were added to the solution in a quantity equal to 3.00% (*w*/*w*) of the final solution. After the addition of microorganisms, the thickener X-Gum was finally added.

### 2.6. Anti-Listeria Activity Determination in Polysaccharides and Protein-Based Films and Coatings: “In Vitro” Studies

The anti-*Listeria* activity of polysaccharides-based and protein-based films and coatings were evaluated against *L. monocytogenes* NCTC 10888. Small samples (2 × 2 cm^2^) of films and an aliquot of the coating solutions containing viable cells of *L. kefiri* were placed onto TSA plates seeded with 10^5^ CFU overnight cultures of *L. monocytogenes* NCTC 10888. The plates were incubated at room temperature and at 4 °C for 24 h and the antagonistic activity was quantified by a clear zone of inhibition in the indicator lawn in contact with the coatings and around the same. The sizes of the inhibition zones (in millimeter) were measured. *L. kefiri* MM5 free polysaccharides-based/proteins-based films and coatings were used as control.

### 2.7. Evaluation of the Antibacterial Activity of Active Films and Coatings in Inoculated Whole Vegetables: “On Food” Studies

The evaluation of the antibacterial activity was determined in individually packaged samples of carrots, zucchini, and tomatoes purchased in the first day of shelf life (indicated by the expiration date on the package). Before starting the study, the microbial contamination of the vegetables was determined. The samples showed on average a microbial load of 50 CFU/g (20 CFU/g for carrots, 70 CFU/g for zucchini and 60 CFU/g for tomatoes, respectively). The presence of *L. monocytogenes* was excluded. On the same day, a pH value was measured on the surface of the vegetables, resulting in pH 6 for carrots and zucchini, and pH 5.5 for tomatoes. The food samples were artificially contaminated by a 5 min immersion in a 10^8^ CFU/mL suspension of *L. monocytogenes* NCTC 10888 in sterile saline solution (NaCl 0.85%). After drying, the contaminated samples were (i) individually packaged in polysaccharide or protein-based films, completely wrapping the sample, and carefully sealing the ends (ii) dipping coated with both types of coatings containing viable cells of *L. kefiri* MM5, and subsequently drying them before starting the experiment. Contaminated samples packed in films and coatings without *L. kefiri* were used as negative controls. All samples were stored at room temperature and 4 °C and, at regular intervals (time 0, 24 h, 3 days, 10 days and 20 days), the viable load of *L. monocytogenes* was determined by direct counting in Palcam agar plates (Oxoid, Milan, Italy). Briefly, 10 g of the samples were collected in sterile plastic bags, added with 90 mL of Buffered Peptone Water (Oxoid, Milan, Italy) and homogenized for 1 min in Stomacher (Stomacher Lab Blender, Seward Medical, London, UK). Serial tenfold dilutions of the obtained suspensions were spread in triplicate on Palcam agar added with selective supplement (Oxoid, Milan, Italy) and plates were incubated aerobically at 37 °C for 48 h. Viable cells of *L. monocytogenes* were enumerated and results expressed as CFU/g. In all negative samples the suspensions were filtered (0.45 mm pore-size filter; Millipore Corp., Bedford, MA) to recover the uncounted *Listeria* cells. Furthermore, MRS agar plates (Oxoid, Milan, Italy) were also used to determine the viable loads of *L. kefiri* in the active protein-based and polysaccharides-based films and coatings containing *Lactobacillus*.

### 2.8. Statistical Analysis

The study was organized using a total of 18 samples for each type of vegetable (3 for each time). To verify the reproducibility of the results, each experiment was replicated three times under the same conditions, and the bacterial count was performed on three plates. The arithmetical mean of the values, expressed as log CFU/mL, was plotted against incubation time. SDs were calculated and error bars reported. The statistical significance was determined by *t*-test, and ANOVA test using statistical program GraphPad Prism 9.2.0. (San Diego, CA, USA). *p*-values were declared significant at ≤0.05.

## 3. Results

### 3.1. Film Characterization

#### 3.1.1. Thickness

The film thickness is reported in Table 1. The polysaccharides film presented a thickness value lower than protein film since the amount of glycerol in protein film is higher than in polysaccharides film. In general, as reported in other studies [39] when the concentration of plasticizer increases, the film thickness also increases.

#### 3.1.2. Barrier Properties

The value of OTR and WVTR measured are reported for each kind of film in Table 2. The values obtained with both kind of films are comparable with values reported in literature for standard polysaccharides-based and proteins-based films [40,41,42,43,44]. The values of permeability obtained for polysaccharides-based films can be defined as very high for OTR and high for WVTR, according to UNI 10534:1995 Italy regulation [45]; instead, the values of permeability obtained with protein film can be defined as high for OTR and medium for WVTR, according to UNI 10534:1995 Italy regulation [45].

The proteins-based and polysaccharides-based films developed in this study present high and very high values of OTR, respectively. In general, this high permeability to oxygen is due to the natural hydrophilicity of proteins and polysaccharides, as macromolecules on the whole [46]. As reported by Wua et al. [47], most edible proteins-based and polysaccharides-based films are excellent barriers against non-polar permeants, such as oxygen and aroma compounds. With regards to WVTR values for polysaccharides-based films, Rhim et al. [48] and Wang et al. [49] refers that the reduction of WVTR observed for alginate films is probably due to the alginate salt, which reduces its permeability through the creation of a tortuous passage to cross the water vapor through the film, increasing the crystallinity of the biopolymer, or decreasing free hydrophilic groups (OH, NH) in biopolymer matrix. On the contrary, protein-based films are in general poor water vapor barriers due to the inherently high hydrophilicity of the proteins and the substantial amount of hydrophilic plasticizers added to the protein-based films.

### 3.2. Anti-Listeria Activity of L. kefiri MM5 Strain and of Active Films and Coatings: “In Vitro” Studies

In the “in vitro” studies, *L. kefiri* MM5 displayed a marked antagonistic activity against *L. monocytogenes* NCTC 10888, with an inhibition zone of about 12 mm, both after incubation at room temperature and at 4 °C (data not shown). The anti-*Listeria* activity was also confirmed when the bacteriocinogenic strain has been incorporated in PoAF, PoAC, PrAF, and PrAC and was revealed by a clear zone of inhibition in the indicator lawn both around the film samples and the coating solutions (Figure 1a,b). No significant differences were found for films and coatings incubated at room temperature or at 4 °C.

### 3.3. Anti-Listeria Activity of Films (PoAF) and Coatings (PoAC) in Inoculated Whole Vegetables: “On Food” Studies

Figure 2a–c reports the mean values of the *L. monocytogenes* viable counts (log CFU/g) detected in the contaminated vegetables, wrapped or coated with treated (live-*L. kefiri* MM5-films and coatings) or untreated (control) materials, and after storage at room temperature. The trend observed for *L. monocytogenes* in carrot samples shows that listeria populations gradually decreased after 72 h both in PoAF and PoAC carrot samples, and a further reduction of about 2.00 log and 3.00 log CFU/g compared to the control was observed after 10 (*p* < 0.01 and *p* < 0.05, respectively) and 20 days (*p* < 0.0008 and *p* < 0.00004), respectively (Figure 2a). As observed in Figure 2b, in the tomato samples packaged with PoAF, *L.monocytogenes* showed a decrease of about 1.00 log over 72 h (*p* < 0.05), compared to the control, followed by a further reduction (about 2.00 log CFU/g) after 10 days (*p* < 0.01). At the end of the study, a reduction of *L. monocytogenes* viable cells of approximately 4.50 log was observed (*p* = 0.0004). In the PoAC tomato samples *L. monocytogenes* viable counts decreased by about 0.50 log over 24 h (*p* < 0.05) and a greater reduction of 3.50 log and 4.00 log, compared to the control (*p* < 0.0001), resulted after 10 and 20 days of storage, respectively. Similar to what observed in the samples of carrots, in both PoAF and PoAC zucchini samples, *L. monocytogenes* population gradually decreased after 72 h, and a further reduction by about 2.50 and 3.50 log CFU/g at the 10th (*p* < 0.05 and *p* < 0.0001, respectively) and 20th days (*p* < 0.0001 and *p* < 0.001, respectively) of storage was obtained (Figure 2c). Lastly, with regard to the LAB viability, we observed over time the growth of *L. kefiri* MM5 in all PoAF and PoAC samples, supporting the evidence of a direct production of the active anti-*Listeria* bacteriocin during its growth and subsequent release on contact surface, as already emerged in the “in vitro” studies (Figure 2d), without significant differences between the two packaging systems. The results for 4 °C are not shown because they were very similar to those obtained at room temperature.

### 3.4. Anti-Listeria Activity of Films (PrAF) and Coatings (PrAC) in Inoculated Whole Vegetables: “On Food” Studies

In the PrAF carrot samples (Figure 3a) the anti-*Listeria* activity was evident after 72 h of storage at room temperature, with a marked reduction in viable counts of about 6.50 log CFU/g over the ten days, compared to the control (*p* < 0.001). On the 20th day, viable *L. monocytogenes* cells were reduced by about 8.00 log CFU/g (*p* < 0.0001). Inoculated PrAC carrot samples showed the best anti-*Listeria* activity over the ten days (*p* < 0.001), followed by a gradual decrease of 4.00 log CFU/g over time (20 days). In the PrAF tomato samples (Figure 3b) *L. monocytogenes* viable counts decreased about 1.00 log and 2.50 log after 24 h and 72 h, respectively, and a further reduction of 5.00 log (*p* < 0.001) was observed at the end of the experiments (20th day). On the tomato samples treated with PrAC an evident anti-*Listeria* activity after 24 h was observed, with a reduction in viable counts of about 2.50 log and a further decrease of 5.50 log CFU/g at the end of the study (*p* < 0.001). On both the PrAF and PrAC zucchini samples (Figure 3c), *L. monocytogenes* population gradually decreased over 72 h, and the best anti-*Listeria* activity was observed over 10 and 20 days, respectively. Notably, the samples packaged with the active protein-based films PrAF showed the best anti-*Listeria* activity of about 8.00 log at the 10th day (*p* < 0.0001), and this reduction in viable counts was maintained until the twentieth day. In the PrAC samples *L. monocytogenes* cells gradually decreased up to the end of the study (20 days) with a reduction in viable counts of about 5.50 log, compared to the control (*p* < 0.0001). Lastly, even in this case, we observed over time the growth of *L. kefiri* MM5 in all PrAF and PrAC samples (Figure 3d), without significant differences between the two packaging systems. The results for 4 °C are not shown because they were very similar to those obtained at room temperature.

## 4. Discussion

Foodborne diseases are a widespread and growing public health and economic problem, but the guarantee of safe foods must also be in accordance with the total respect of nature. The aim of this study was to identify a natural and ecological approach to control the survival and/or growth of *L. monocytogenes* on whole and intact plant surfaces. New edible films and coatings containing a probiotic strain capable of growing, proliferating, and producing an anti-*Listeria* bacteriocin have been developed. The potential use of bacteriocin-producing LAB, both as bioprotective and probiotic agents, has recently received increased attention by the scientific community. They are capable to reduce the risk of food-related diseases by competing within complex microbial communities and positively influencing the health of the host [50,51]. Consumers are increasingly focused on food products that not only meet their nutritional demands, but also have additional benefits improving their health. For this purpose, a LAB bacteriocin producer was added to films and coatings of different compositions, resulting in bioactive packaging solutions with high biological value, due to the presence of the probiotic LAB. The gradual growth of the probiotic strain *L. kefiri* MM5 inside both types of edible films and coatings, without an initial decrease in vital load, as reported in other studies [52,53], underlines that the process conditions haven’t had an impact on bacterial survival. The storage temperature (4 °C) hasn’t also affected the probiotic survival, contrary to what has been pointed out by other authors [54]. Both proteins-based and polysaccharides-based films and coatings showed a good anti-*Listeria* activity, notably over 10 and 20 days of experimentation. About the polysaccharides-based packaging, the coatings showed better antibacterial activity than coatings. This is probably due to its higher amount of sodium alginate (6.00% *w*/*v*) compared to the films (2.00% *w*/*v*). In fact, although sodium alginate coatings or films show little intrinsic antimicrobial activity, the addition of antibacterial substances or lactic bacteria enhances this feature [55].

On the same samples, protein-based packaging displayed the best antibacterial activity against *L. monocytogenes* (*p* < 0.0001) over 10 days of storage. As reported in other investigations, the protein matrix could be more effective in maintaining LAB viability and activity (the last intended as secretion of antibacterial substances like bacteriocins) than the polysaccharides one [56]. The differences in oxygen permeability protein-based and polysaccharides-based matrices may also play a role in the LAB survival [57]. Another factor affecting the antibacterial activity shown by proteins-based packaging could be the better solubility, due to an increased water vapor permeability values and, consequently, to an improved solubility of protein-based films and coatings. As reported in other studies, this characteristic could be directly correlated with a better release of the incorporated bioactive compounds [58,59,60]. In our study, protein-based films showed the better anti-*Listeria* activity, compared to coatings; these last ones were prepared by adding X-Gum as thickener that decreased the water vapor permeability values, and consequently, the solubility of the protein-based coatings and a less release of bioactive compounds.

Another advantage in using protein-based materials is the biodegradability. The research for new, unconventional materials that can be used for packaging food in a more sustainable and eco-friendly way has focused on the use of food waste. Agro-industrial residual biomass deriving from the entire food chain are by far those generating yearly the largest amount of co-products, residues, and wastes. According to the Food and Agriculture Organization of the United Nations (FAO), about 1.30 billion tons of food is lost or wasted per year, along the whole production chain starting at the production stage and ending at the consumer level. The global volume of food wastage goes up to 1.6 billion tonnes when inedible parts are also included [61]. The generation of byproducts from food processing is inevitable and their disposal is, nowadays, one of the major challenges. About 38% of the residue wastes are generated during food processing [62]. Among all food commodities, fruit and vegetables (such as legumes) are the largest food waste contributor, representing 44.00% of the global food waste, roots and tubers contributing by 20.00%, and cereal by 19.00% [63]. Food waste constitutes a largely under-exploited residue from which a variety of valuable chemicals can be derived. In fact, those residues are yet a rich source of useful compounds like carbohydrates, proteins, lipids, and polyphenols, which could find high-value applications in many industrial sectors [64]. Specifically, legume residues are rich in proteins and peptides as well as fibers that can be extracted and further valorized [65,66]. The proteins used in this study were extracted from peas by-products using a method of extraction developed in a SSICA patent (N. Patent. 0001399500) [30], an environmentally friendly method, since it does not use organic solvent and high temperature treatments and it operates in neutral conditions. Specifically in the European FP7 project LEGUVAL [67], a simple and sustainable method for extracting legume and fibers from legume by-products, achieving high protein percentages in the final extracts was set up and developed. The method was applied to different kinds of legumes by-products at laboratory scale, namely, peas, lentils, and beans. Among all legumes by-products tested, pea by-products were selected due to the simplicity of matrix, good processability, availability, and high quantitative yield of extraction. In fact, with this innovative method to extract proteins from such by-products, it’s possible to obtain a final protein extract with a purity degree close to 80.00%. Proteins can be considered an optimum basis for developing packaging biodegradable films due to their relative abundance, biodegradability, good film-forming ability, and high nutritional value [42]. Developments have been observed in the European FP7 LEGUVAL [67] and BBI PROLIFIC [68] projects, for example, where proteins derived from legumes by-products are valorized for the production of film and coating to be applied on plastic multilayer system packaging with improved barrier properties and as food ingredients and feed additives.

## 5. Conclusions

The active packaging described in this study can ensure a long-lasting antibacterial activity in contact with food, preserving its microbiological quality, with a consequent increase in the shelf-life, compared to the addition of pre-formed bacteriocins or other natural substances. Moreover, the development of active packaging containing live probiotics not only will keep food from pathogens and spoilage bacteria, but will also provide beneficial effects on the gastrointestinal tract of consumers. Lastly, to create safe, innovative, and advanced green packaging it becomes essential to both use natural compounds and to apply sustainable principles and methods aimed not to spoil natural resources and the environment. The active films and coatings proposed were obtained using edible ingredients, easily available, at low cost and environmental impact (green). This approach will allow the reduction of chemicals and synthetic plastics used for food preservation, notably in highly perishable fresh products, such as vegetables. In conclusion, these probiotic films and coatings obtained at low cost are green and feasible alternatives for controlling foodborne pathogens, improving food safety, and providing health benefits for the consumers.

## Figures and Tables

**Figure 1 foods-11-02632-f001:**
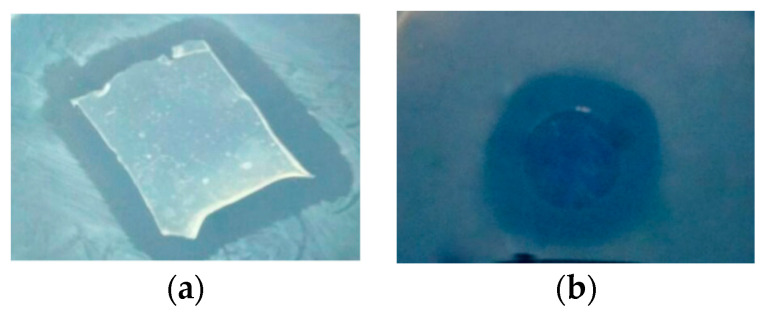
Example of anti-*Listeria* activity, detected by agar diffusion assay. After incubation at room temperature for 24 h, the antagonistic activity was quantified by a clear zone of inhibition in the indicator lawn around both active film (**a**) and coating (**b**).

**Figure 2 foods-11-02632-f002:**
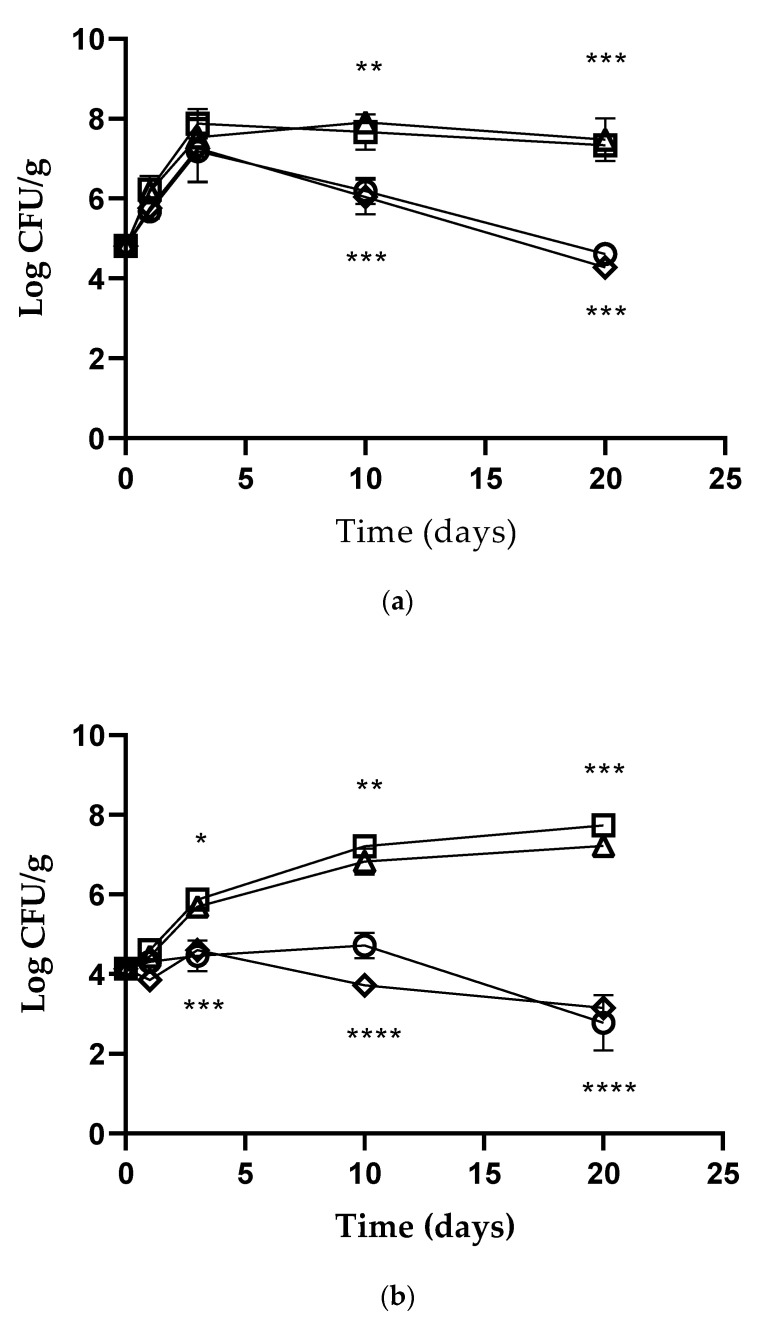
*Listeria monocytogenes* NCTC 10888 viable counts (log CFU/g) observed in the contaminated (**a**) carrots, (**b**) tomatoes, and (**c**) zucchini samples packaged in PoAF [🞅], PoAC [♢], PoF control [△] and PoC control [☐]. *Lactobacillus kefiri* (**d**) viable counts (log CFU/g) observed in the contaminated carrots, tomatoes, and zucchini samples packaged in PoAF [🞅, △, 
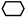
], PoAC [☐, ♢, ●]. Error bars represent standard deviations. *p*-values of < 0.05 (*), *p* < 0.01 (**), *p* < 0.001 (***) and *p* <0.0001 (****) were considered significant by *t*-test and ANOVA. ns stands for not statistically significant.

**Figure 3 foods-11-02632-f003:**
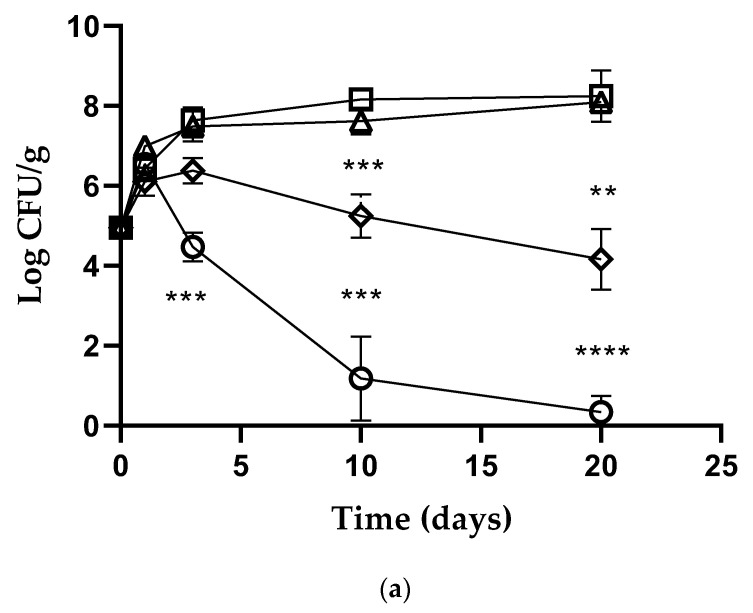
*Listeria monocytogenes* NCTC 10888 viable counts (log CFU/g) observed in the contaminated (**a**) carrots, (**b**) tomatoes, and (**c**) zucchini samples packaged in PrAF [🞅], PrAC [♢], PrF control [△] and PrC control [☐]. *Lactobacillus kefiri* (**d**) viable counts (log CFU/g) observed in the contaminated carrots, tomatoes, and zucchini samples packaged in PrAF [🞅, △, 
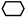
], PrAC [☐, ♢, ●]. Error bars represent standard deviations. *p*-values of < 0.05 (*), *p* < 0.01 (**), *p* < 0.001 (***) and *p* < 0.0001 (****) were considered significant by *t*-test and ANOVA. ns stands for not statistically significant.

**Table 1 foods-11-02632-t001:** Thickness of films, the results are reported as the mean ± sd.

Sample	Thickness (µm)
Polysaccharides film	89.10 ± 11.10
Protein film	160.70 ± 14.10

**Table 2 foods-11-02632-t002:** Barrier properties of films, the results are reported as the mean ± sd.

Sample	Test Conditions	WVTR [g/(m^2^ 24 h)]	OTR [cm^3^/(m^2^ 24 h)]
Polysaccharides film	T 23 °C RH 10%	1.73 ± 0.07	0.10 ± 0.02
Protein film		16.46 ± 2.00	2.90 ± 0.13

## Data Availability

Not applicable.

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
