# Peer review of "Eco-Friendly Edible Packaging Systems Based on Live-Lactobacillus kefiri MM5 for the Control of Listeria monocytogenes in Fresh Vegetables"

_foods, 2022, doi:10.3390/foods11172632_

Round 1
Reviewer 1 Report
-Authors need to adapt the formatting of the text to the Journal's Rules, and also check the citations of the references, some bring the year of publication in addition to the reference number (this form of the Journal's Rules).
-The text is not organized into paragraphs, and it gets a little confusing, especially in some sentences that are very long. The discussion is quite confusing and disorganized.
- The authors should cite the references used to prepare the films.
- It is very important to carry out the statistical analysis of all results, for several results presented there is no statistical analysis.
- Figures should have better resolution and larger letter sizes.
Author Response
Dear Reviewer, thanks for the right and useful comments. I hope that the answers to the questions and the parts added in the text comply with what you have requested.
Yours sincerely

Reviewer 2 Report
Comments and and Suggestions for Authors:
- Please provide a short description of coating method or packaging by films for three types of vegetable samples.
- Photography of the treated tomatoes, carrots and zucchini samples (after coating or packaging by PoAF, PoAC) would be interesting to include.
- It is not clear the experimental design used for this work. How many replicates per treatment were used. How many tomatoes per treatment were used? How many tomatoes were sampled for analysis each time?
- In discussion part: authors need to explain why “protein-based film resulted more active towards the pathogen than protein-based coating.” Similar to polysaccharide - based films and polysaccharide - based coating”.
- Typing mistake: "besed films" in some sentences.
Author Response

(The authors gave the same response as above.)

Reviewer 3 Report
Overall, the authors used inappropriate terms throughout the manuscript and the manuscript was poorly written which makes the manuscript cumbersome to read. In addition, the manuscript suffers from grammatical errors that sometimes lead to misleading information. For example, in Section 2.2, instead of “added” the authors can use a more technical term such as “spreaded”. The sequence of methodology and results section does not jive, which makes it difficult for the reader to follow through. Some of the results were not justified/discussed in the discussion section.
The title is general, the author can directly mention what kind/type of food bio-preservation was used.
The manuscript's main ideas were not separated into paragraphs, making it difficult to highlight important information.
In the introduction part, the authors mentioned “inhibit or kill other related or unrelated microorganisms”. Please specify related or unrelated microorganisms.
Why the use of these 3 vegetables and what is the relationship to Listeria monocytogenes contamination. Are these 3 vegetables the main concern in LM contamination? If yes, this needs to be justified and proven in the introduction section.
What is the procedure following immersion? Did the author pack the vegetables immediately? Before packaging, was there any drainage or how did the author remove excess suspension?
Figure 1 is a bit confusing and not self-explanatory. In addition, the graphs are not clear.
What will be the significance of having to measure the thickness for the material? This was not discussed in the discussion section.
Author Response

(The authors gave the same response as above.)

Round 2
Reviewer 3 Report
1. Please label Figure 1a and 1b. Add details on what the author means by ant-listeria activity retention. What are those? The figure needs to be self-explanatory.
Please add punctuation (where applicable/appropriate) throughout the manuscript. Eg: To create innovative and advanced green packaging, ....
Author Response
Thanks again for the comments. The suggested changes have been made
